# The Al Doping Effect on Epitaxial (In,Mn)As Dilute Magnetic Semiconductors Prepared by Ion Implantation and Pulsed Laser Melting

**DOI:** 10.3390/ma14154138

**Published:** 2021-07-25

**Authors:** Ye Yuan, Yufang Xie, Ning Yuan, Mao Wang, René Heller, Ulrich Kentsch, Tianrui Zhai, Xiaolei Wang

**Affiliations:** 1Songshan Lake Materials Laboratory, Dongguan 523808, China; 2Helmholtz-Zentrum Dresden-Rossendorf, Institute of Ion Beam Physics and Materials Research, Bautzner Landstrasse 400, 01328 Dresden, Germany; y.xie@hzdr.de (Y.X.); n.yuan@hzdr.de (N.Y.); m.wang@hzdr.de (M.W.); r.heller@hzdr.de (R.H.); U.Kentsch@hzdr.de (U.K.); 3College of Physics and Optoelectronics, Faculty of Science, Beijing University of Technology, Beijing 100124, China; trzhai@bjut.edu.cn

**Keywords:** magnetic semiconductors, ion implantation, co-doping, magnetic properties

## Abstract

One of the most attractive characteristics of diluted ferromagnetic semiconductors is the possibility to modulate their electronic and ferromagnetic properties, coupled by itinerant holes through various means. A prominent example is the modification of Curie temperature and magnetic anisotropy by ion implantation and pulsed laser melting in III–V diluted magnetic semiconductors. In this study, to the best of our knowledge, we performed, for the first time, the co-doping of (In,Mn)As diluted magnetic semiconductors by Al by co-implantation subsequently combined with a pulsed laser annealing technique. Additionally, the structural and magnetic properties were systematically investigated by gradually raising the Al implantation fluence. Unexpectedly, under a well-preserved epitaxial structure, all samples presented weaken Curie temperature, magnetization, as well as uniaxial magnetic anisotropies when more aluminum was involved. Such a phenomenon is probably due to enhanced carrier localization introduced by Al or the suppression of substitutional Mn atoms.

## 1. Introduction

III-Mn-V diluted ferromagnetic semiconductors (DFSs) have received a great deal of attention because of their enormous potential for spintronic application [1,2] due to their characteristic that electronic and ferromagnetic properties are coupled by itinerant holes [3,4,5,6,7,8]. However, because the ultra-low solubility of Mn in III–V semiconductors is significantly (normally several orders of magnitude) below the threshold value of appearing ferromagnetism, the preparation of epitaxial DFS film is extremely challenged. Even for the most canonical DFS (Ga,Mn)As, mostly low-temperature molecular beam epitaxy (LT-MBE) has been employed for its epitaxial film preparation [4,7,8], which significantly prohibited the development of DFSs. Moreover, such an obstacle threatens the most conventional technique of semiconductor modification, i.e., co-doping which is productive to present new physics [9,10,11,12,13,14]. However, ion implantation followed by the pulsed laser melting method provides an alternative solution to this problem due to its second non-equilibrium grown essence, and therefore, it has been used in several hyper-doping cases in which some are even impossible to approach by LT-MBE [15,16,17]. Thus, it is important to explore the co-hyper-doping effect in various DFSs by ion implantation and pulsed laser melting (PLM).

According to the *p-d* Zenner model [3,5], Curie temperature and magnetization of DFS are strongly commensurate with *p-d* hybridization, and such *p-d* coupling is, in principle, more intensive between Al and As than between Ga and As. Accordingly, (Al,Mn)As is theoretically expected to exhibit higher *T*_C_ than (Ga,Mn)As at the same Mn concentration and hole density [3]. Unfortunately, such an expectation has not been subsequently observed in LT-MBE grown Al-doped (Ga,Mn)As sample [13,18], in which only magnetic anisotropy switching happens, and such behavior is explained by the enhanced hole localization introduced by Al doping. It is worth while noting that the absence of an apparent increase in *T*_C_ is probably due to the stronger *p-d* hybridization between Al and As, which enables the system to be more insulating and also reduces magnetization. Therefore, it seems that the *T*_C_ modification is a trade-off between *p-d* coupling enhancement (increase *T*_C_) and *p-d* coupling enhancement induced by carrier localization (decrease *T*_C_) in various DFS candidates [13,18,19,20]. On the one hand, it is important to study the Al-doped (In,Mn)As, since the *p-d* coupling between Al and As is much stronger than that between In and As [3,5,21]; however, InAs presents a higher hole mobility [22,23], and accordingly, a high *T*_C_ is expected in Al-doped (In,Mn)As. On the other hand, due to the challenged preparation of epitaxial (In,Mn)As DFSs, studies on co-doping effect in (In,Mn)As are limited and new phenomena should be investigated.

In this study, Al-co-doped (In,Mn)As samples were realized by ion implantation combined with a pulsed laser melting technique, and by gradually increasing the Al implantation fluences, the Al doping concentration was accordingly raised. According to the results of Rutherford backscattering spectrometry and ion channeling (RBS/C), all (In,Mn)As and Al-co-doped (In,Mn)As films display epitaxial structure, which is comparable with the virgin InAs wafer. Upon raising the Al content, magnetization and *T*_C_ are both apparently reduced together with weakened out-of-plane uniaxial magnetic anisotropy, which is probably due to the enhanced localization or reduced substitutional Mn atoms. Similar phenomena have been observed in our previous studies on co-doped (Ga,Mn)As and (Ga,Mn)P samples.

## 2. Materials and Methods

The Al-doped (In,Mn)As samples were prepared by ion implantation combined with the PLM technique. Before the Al implantation, all InAs substrates were implanted with Mn ions to achieve the Mn doping. Mn and Al implantation were both performed at an angle of 7° to avoid channeling effect. For Mn doping, the implantation fluence was set as 2.4 × 10^16^/cm^2^, and the implantation energy was 100 keV at room temperature. Afterwards, Al implantation was carried out on post Mn implanted samples with fluences of 1.7 × 10^16^/cm^2^ and 3.4 × 10^16^/cm^2^ at an energy value of 60 keV at room temperature, and the samples were referred to as InMnAlAs-4 and InMnAlAs-8, respectively. One Mn implanted sample was selected as the reference for a comparison, i.e., InMnAlAs-0. After implantation, the doped region became amorphous, and thus recrystallization was necessary. For the annealing process, a UV pulsed laser with nanosecond pulse was employed for the recrystallization. The pulsed laser treatment with a 28 ns duration was performed in air atmosphere, and an energy of 0.2 J/cm^2^ was selected, which is the optimal annealing condition for InAs recrystallization [21,24]. The wavelength of the pulse laser is 308 nm. Magnetic properties were explored by a Superconducting Quantum Interference Device vibrating sample magnetometer (SQUID-VSM, Quantum Design, US) equipped with a low field option. For the temperature dependent thermo-remanent magnetization (TRM) measurement, first, the sample was cooled from 300 to 5 K under a 5 kOe magnetic field for magnetic saturation. Afterwards, the field was totally removed by magnet reset operation, and then the warm process started; meanwhile, the data collection began.

## 3. Results and Discussion

### 3.1. SRIM Calculation

Figure 1 displays the results of Mn and Al distribution in the InAs matrix by the stopping and range of ions in matter (SRIM) calculations [25]. As shown in the figure, both Mn and Al present a Gaussian distribution at a depth of 200 nm. Interestingly, even the implanting energy of Mn (100 keV) is much larger than that of Al (60 keV), and the Mn maximal peak appears at a shallower depth (54 nm) than that of Al (67 nm), due to the different stopping abilities of InAs lattice to these two elements. According to the SRIM simulation, the longitudinal straggling for Mn and Zn in the InAs matrix is 37.5 and 53 nm, respectively, thus, the thickness of the Mn- and Zn-doped regions can be treated as 75 and 106 nm by 2∆R_P_, respectively. The overlap of the doped regions indicates a valid co-doping effect. When the atomic density of InAs is considered, the atomic ratio of implanted atoms to indium atoms is easily obtained. As a result, the maximal Mn concentration is 14.7% when the implanting fluence is 2.4 × 10^16^/cm^2^, and the average doping concentration is around 12%. However, according to our previous study [24], the only-Mn-doped sample exhibited an average Mn concentration of 8.7% in the doped region, due to the diffusion of Mn caused by pulsed laser-induced liquid phase epitaxy. For Al implantation, the average concentrations are calculated as 6% and 12%, respectively, when the implantation fluences are selected as 1.7 × 10^16^/cm^2^ and 3.4 × 10^16^/cm^2^.

### 3.2. Structure of All Co-Doped Samples

For recrystallizing as-implanted samples, a pulse (28 ns) UV laser was employed to activate the implanted dopants. During the pulsed laser melting treatment, most laser energy is absorbed by the amorphized implanted region. The absorbed energy is so high that the whole implanted region begins to melt, whereas the InAs substrate remains at room temperature. As a result, the huge temperature gradient drives ultra-fast recrystallization at a speed of several m/s and the so-called liquid epitaxy process happens. Due to the ultra-fast growth process, implanted Mn and Al atoms have no time to perform long-range diffusion and are driven into the matrix lattice.

To reveal the structure of Mn- and Al-co-doped and Mn-doped InAs samples, Raman spectroscopy was used. The spectra are shown in Figure 2. All samples present two obvious peaks at wavenumbers of 225 and 233 cm^−1^. For the InMnAlAs-0 sample, a broadened signal and a sharp LO signal are both observed, indicating that both vibration modes are detectable from the [001] direction. However, according to the selection rule of the zinc-blende structure, only the longitudinal optical (LO) phonon mode is allowed in the backscattering configuration, whereas the transverse optical (TO) phonon mode is forbidden. Therefore, it is not possible that the peak at 225 cm^−1^ is original from the TO mode. Actually, due to the presence of itinerant holes inside the Mn-doped region, part of the LO signals transfer into coupled plasmon-LO-phonon mode (CPLOM) which is present at a wavelength of 225 cm^−1^ between LO vibration and itinerant holes [26]. However, the introduction of Al atoms into the (In,Mn)As and the enhanced carry localization by Al involvement largely eliminate coupled contribution by free holes [27,28]. Therefore, the CPLOM signal is largely suppressed in the InMnAlAs-4 sample as compared with in the InMnAlAs-0 sample. However, the continuous increase of Al concentration increases the lattice disorder in the matrix, which leads to an intensive broadening of the vibration peak.

In addition to Raman microscopy, the structure of all co-doped samples together with the reference (In,Mn)As and virgin InAs samples are investigated by RBS/C. THE results are shown in Figure 3. According to the random spectra, indium and arsenic signals are both pronounced; however, the Mn and Al signals are much weaker, even invisible, due to a low concentration of only several percentages and the overlapping with the strong signal from In and As. After normalization, it is possible to evaluate the crystalline quality by the ratio of backscattering yields obtained under the channeling conditions and the random beam incidence, which is defined as χ_min_. As shown in Figure 3, upon increasing Al doping density, the χ_min_ only reveals a slight increase, from 9.8%—via 11.0%, 11.2%, and finally to 13.3%—indicating that crystallization deviates slightly from the InAs lattice, which is caused by Mn and Al doping. However, the epitaxial nature of the co-doped films persists, which excludes the contribution from amorphization or polycrystaliziation to the latter discussed manipulation of the magnetic properties.

### 3.3. Magnetic Properties

During the gradually increasing Al implantation fluences, the evolvement of magnetic properties is investigated, and one (In,Mn)As without Al doping is present as a reference. Figure 4a shows the magnetic field dependent magnetization after subtracting the diamagnetic signal from InAs substrates. The InMnAlAs-0 sample presents a typical ferromagnetic feature, i.e., highly square-like hysteresis loop with a low coercivity field (107 Oe) and a magnetization of ~23 emu/cm^3^. According to Figure 4b, the sample shows a high *T*_C_ of 80 K where the magnetization vanishes [24]. Moreover, the *M-T* curve shows a concave-like shape, and this is reminiscent of mean-field theory approximation [5,24,29]. The magnetic field and temperature dependent magnetization both unambiguously prove the ferromagnetic nature of our virgin (In,Mn)As. Unexpectedly, the Al doping directly results in reducing magnetization and enhancing coercivity; the saturation magnetization gradually decreases from 23.5, via 19.8, finally to 15.7 emu/cm^3^; meanwhile, the coercivity contrarily rises from 107 via 674 to 993 Oe. Additionally, the thermo-remanent magnetization at 5 K decreases from 23.0, via 20.3, finally to 14.1 emu/cm^3^. Actually, the suppression of magnetization and Curie temperature induced by Al doping has been observed in a series of (Ga,Mn)As samples [13]; in a study by [3], there were two different mechanisms proposed: (i) part of Mn atoms are driven into the interstitial sites; and (ii) the localization of carriers is enhanced by Al alloying due to stronger *p-d* exchange coupling However, in the low Al concentration doped region (<20%), the first function worked dominantly, and a similar phenomenon appeared in Zn-doped (Ga,Mn)As according to our previous research [14]. This is the most probable explanation for the Al-doped (In,Mn)As case, since, for the highest Al concentration doped sample, it is observed that in addition to the reduction in Curie temperature, the *M–T* curve deviates from the mean-field approximation, and such a phenomenon is in good agreement with a decrease in substitutional Mn concentration [24]. Unfortunately, it is extremely difficult to examine the effect of enhanced localization [24,25] in our sample, because of the conducting InAs substrate.

According to an XMCD (X-ray magnetic circular dichroism) study on Al-doped (Ga,Mn)As by Edmonds et al. [27], the doping of aluminum modulates the strain state and further leads to switching of the uniaxial magnetic anisotropy. Thus, it is worthwhile investigating the changing of magnetic anisotropy upon increasing the concentration of aluminum in our (In,Mn)As samples. The results are shown in Figure 5. However, in our samples, although all samples present typical uniaxial magnetic anisotropy with an out-of-plane magnetic easy axis, the anisotropy feature presents a changing behavior with increasing Al doping concentration. As shown in Figure 5a, the InMnAlAs-0 sample exhibits a specific perpendicular uniaxial magnetic anisotropy; the *M–H* curve is highly square-like when the magnetic field is applied along the out-of-plane direction, while a much higher saturation field (2.5 kOe) and lower remanent magnetization are both observed along the in-plane direction. Such a phenomenon duplicates the same characteristics in previous (In,Mn)As prepared by both ion implantation and LT-MBE [6,24].

However, when the doped Al concentration is increased, the *M–H* hysteresis loops of the out-of-plane direction gradually change from a square shape, while the *M–H* curves along the in-plane direction start to open a loop. Such a transform can be quantitatively evaluated by the anisotropy constant. Calculated from the difference of integral area of *M–H* loops between out-of-plane and in-plane directions, the anisotropy constants are 2.71 × 10^4^, 2.13 × 10^4^, and 1.60 × 10^4^ erg/cm^3^ for the InMnAlAs-0, InMnAlAs-4, and InMnAlAs-8 samples, respectively. There is a decline in the anisotropy constants; however, there is no change in the uniaxial magnetic anisotropy as He irradiated GaMnAsP [30]. For a comparison, the magnetic properties of all samples are listed in Table 1.

## 4. Conclusions

In summary, we firstly report the co-doping effect of (In,Mn)As by Al through co-implantation combined with a pulsed laser melting technique. Upon increasing the Al concentration by raising implantation fluences, all doped samples present well-preserved epitaxial structures, and the magnetic properties are tuned. The Curie temperature and magnetization are reduced together with modifications of the anisotropy constants. Unexpectedly, the above-mentioned phenomena are not induced by the amorphization, which is confirmed by the RBS channeling spectra, and it is possible to explain the phenomenon by the enhanced carrier localization or the decreased substitutional Mn atoms introduced by aluminum incorporation.

## Figures and Tables

**Figure 1 materials-14-04138-f001:**
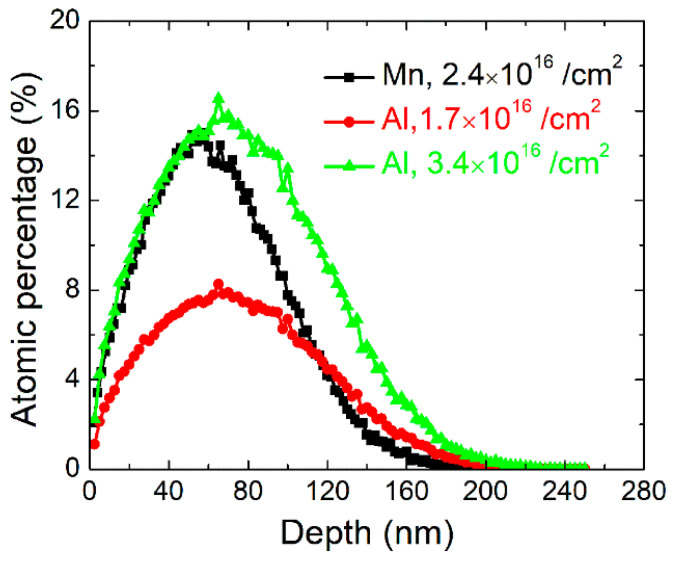
SRIM calculation of implanted Mn and Al distribution in InAs matrix.

**Figure 2 materials-14-04138-f002:**
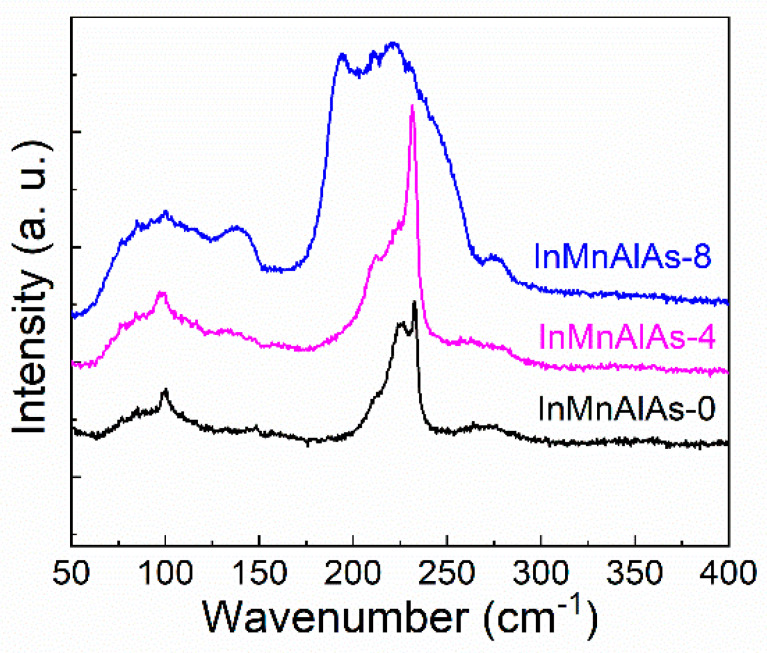
Raman spectra of samples InMnAlAs-0 (black); InMnAlAs-4 pink); InMnAlAs-8 (blue).

**Figure 3 materials-14-04138-f003:**
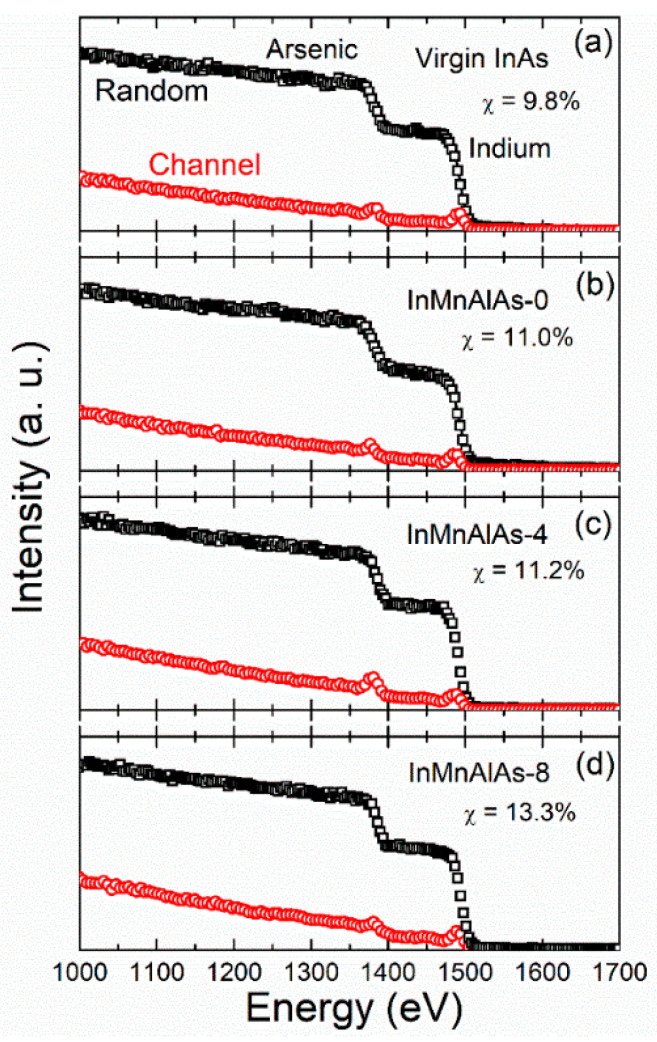
The RBS random (open squares) and channeling (open circles) spectra of samples: (**a**) virgin InAs; (**b**) InMnAlAs-0; (**c**) InMnAlAs-4; (**d**) InMnAlAs-8.

**Figure 4 materials-14-04138-f004:**
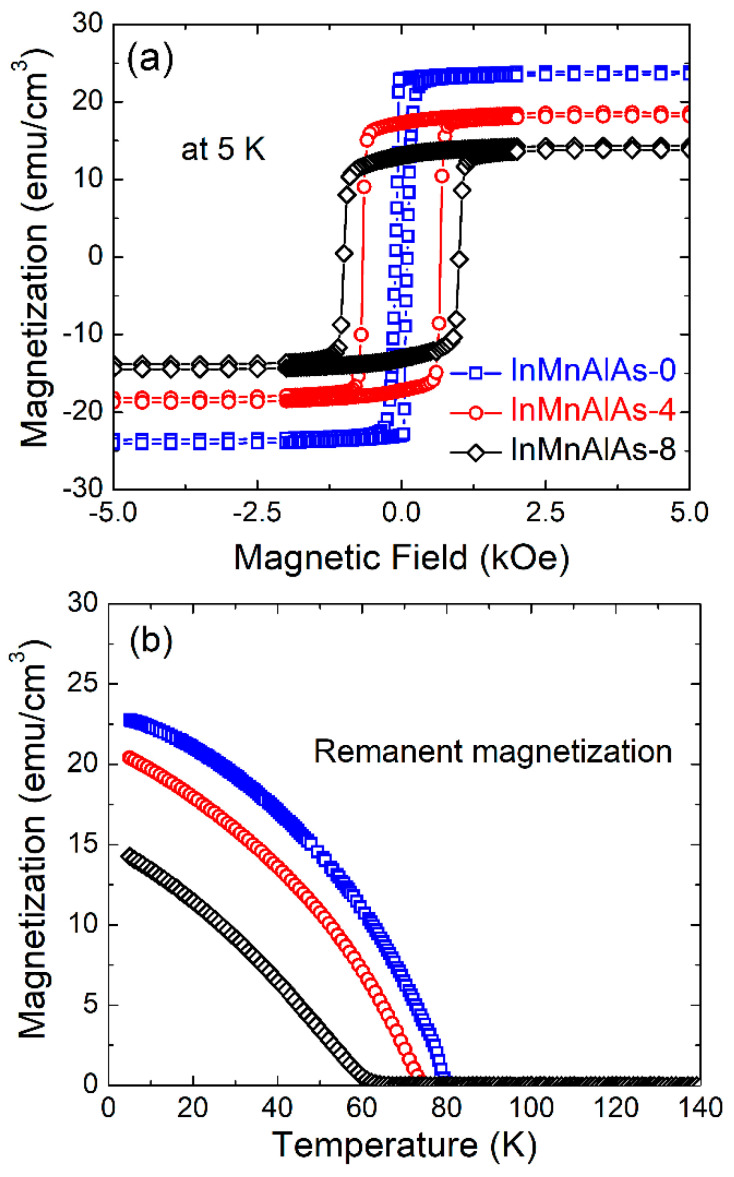
(**a**) Magnetic field dependent magnetization of the InMnAlAs-0 (squares), InMnAlAs-4 (circles), and InMnAlAs-8 (diamonds) samples, at 5 K, when the magnetic field is applied along the out-of-plane direction; (**b**) temperature dependent remanent magnetization at zero field of three samples.

**Figure 5 materials-14-04138-f005:**
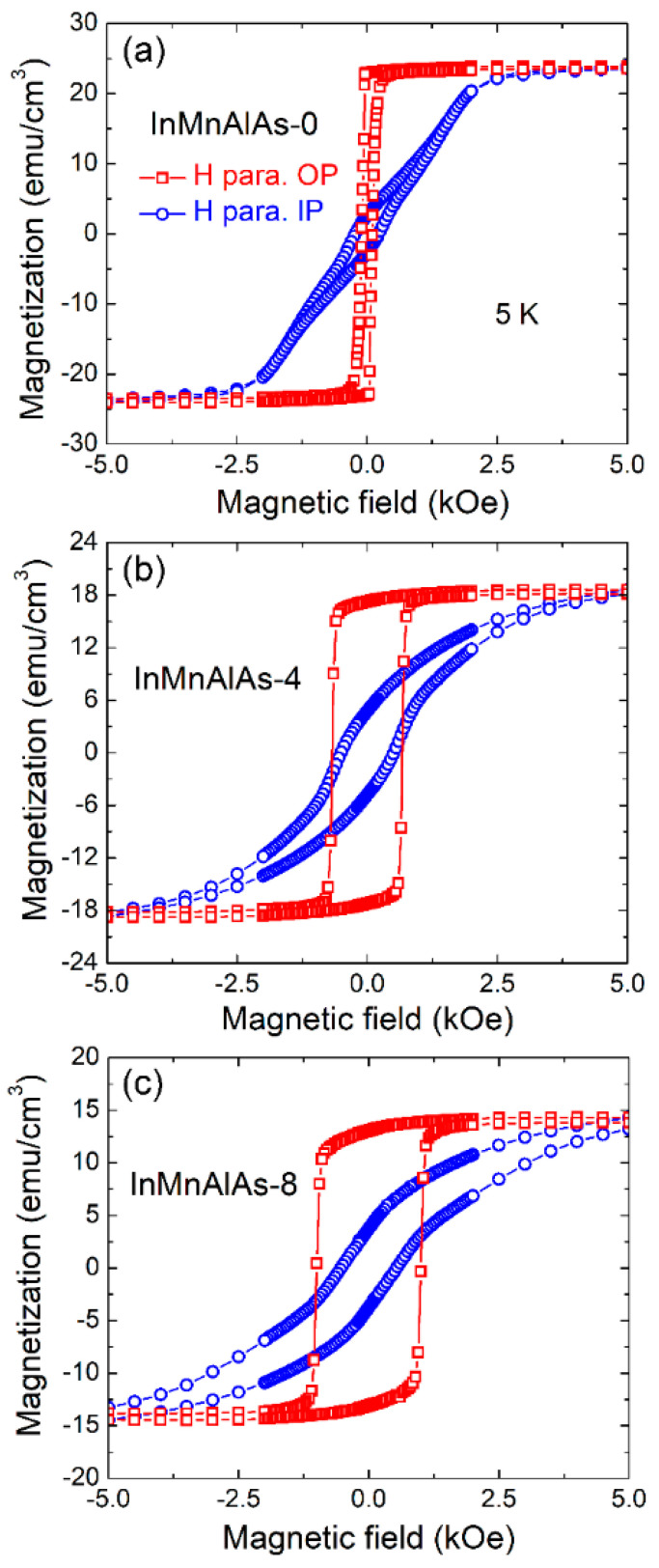
The magnetic field dependent magnetization of samples: (**a**) InMnAlAs-0; (**b**) InMnAlAs-4; (**c**) InMnAlAs-8, when the magnetic field is applied parallel (circles) and perpendicular (squares) with the plane.

**Table 1 materials-14-04138-t001:** Al implantation fluences, remanent magnetization, Curie temperature, coercivity, and anisotropy constant for the InMnAlAs-0, InMnAlAs-4, and andInMnAlAs-8 samples.

Sample No.	Al Imp. Flu. (/cm^2^)	*M*_R_ (emu/cm^3^)	*T*_C_ (K)	*H*_C_ (Oe)	*K* (10^4^ erg/cm^3^)
InMnAlAs-0	0	22.7	80	107	2.71
InMnAlAs-4	1.7 × 10^16^	20.3	72	674	2.13
InMnAlAs-8	3.4 × 10^16^	14.3	60	993	1.60

## Data Availability

The data presented in this study are available in the article.

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
