# Peer review of "The Al Doping Effect on Epitaxial (In,Mn)As Dilute Magnetic Semiconductors Prepared by Ion Implantation and Pulsed Laser Melting"

_materials, 2021, doi:10.3390/ma14154138_

Round 1
Reviewer 1 Report
This manuscript introduces the effect of Al doping by the co-implantation combined with subsequent pulsed laser annealing technique for modulating its electronic and ferromagnetic properties.
However, this manuscript lacks originality.
Also, the experimental conditions of the pulse laser melting as one major technology are not described in detail. If the pulse laser melting technology is an important factor in the results, the author should explain in detail the conditions under which the pulselaser melting technology was applied, and the “Results and Discussion” section also needs explanation.
In line 44, the word “PLM” appears for the first time. Full name is required.
Author Response
Comment 1: this manuscript lacks originality.
Answer: Thanks for suggestions from the referee. However, for the comment of “this manuscript lacks originality.”, it is not objective because such work is really the first successful preparation of Al doped (In,Mn)As DMS. In spite that the method of ion implantation combined with pulsed laser melting has been employed in achieving dilute ferromagnetic semiconductors, the Al co-doped (In,Mn)As has not been achieved due to the intrinsic challenge of (In,Mn)As epitaxy by conventional LT-MBE. Therefore, this work exhibits enough originality.
Comment 2 Also, the experimental conditions of the pulse laser melting as one major technology are not described in detail. If the pulse laser melting technology is an important factor in the results, the author should explain in detail the conditions under which the pulse laser melting technology was applied, and the “Results and Discussion” section also needs explanation.
Answer: Thanks for pointing this, actually the very detailed information of laser treatment has been supplied in Materials and methods part in page 2 as below:
“For the annealing process, a UV pulsed laser with nano-second pulse was employed for the recrystallization. The pulsed laser treatment with a 28 ns duration is performed in air atmosphere, and an energy of 0.2 J/cm2 was selected which is the optimal annealing condition for InAs recrystallization [19, 22].”
But for sure, one more sentence about the used laser annealing condition is added:
“The wavelength of the pulse laser is 308 nm.”
More detailed discussion about the laser treatment is added in Section 3.2 in the corrected version:
“For recrystallizing as-implanted samples, the pulse (28 ns) UV laser was employed to activate the implanted dopants. During the pulse laser treatment, the main part of laser energy is absorbed by amorphized implanted region. The absorbed energy is so high that the whole implanted region becomes melting, whereas the InAs substrate still keeps room-temperature. As a result, the huge temperature gradient drives ultra-fast recrystallization at a speed of several m/s and the so-called liquid epitaxy process happens. Due to the ultra-fast growth process, implanted Mn and Al atoms have no time to perform long-range diffusion and are driven into the matrix lattice.”
Comment 3: In line 44, the word “PLM” appears for the first time. Full name is required.
Answer: Thanks for pointing this. We add the full name in the revised manuscript.
Reviewer 2 Report
Authors report on The Al doping effect on epitaxial (In,Mn)As dilute magnetic semiconductors prepared by ion implantation and pulse laser melting. The paper is averagely written and organized but still contains multiple grammatical and typo mistakes (inclusion in the title).
In the introduction section, authors failed to highlight clearly the novelty of their work in the context of similar ones, especially that this topic is heavily investigated in the literature.
The paper is unbalanced between materials characterizations and performance.
The method section should be detailed in such a way that one could reproduce these results with fidelity.
The references list is not updated with the most recent and most relevant work in the field.
What is the results fluctuation? This should be detailed in the paper. This is mandatory point.
Discussion of results is weakly supported but data.
I believe that the iteration here provided does not constitute any novelty justifying its publication in this journal, especially with the fact that this topic is heavily investigated in the relevant literature. This work is still needing an in-depth improvement to be considered for publication in an academic journal.
Author Response
Comment 1: In the introduction section, authors failed to highlight clearly the novelty of their work in the context of similar ones, especially that this topic is heavily investigated in the literature.
Answer: Although the preparation of DMSs by ion implantation and pulse laser melting has been performed and studied, the investigation on Al co-doped (In,Mn)As has not been carried out. In particularly, the importance and background of the current work has been shown in Introduction part:
“It will be of great meaning to study the Al doped (In,Mn)As, since the p-d coupling between Al and As is much stronger than that between In and As [1, 3, 19], however InAs presents a higher hole mobility [20, 21]. Accordingly, a high TC is expected in Al doped (In,Mn)As. On the other hand, due to the challenged preparation of epitaxial (In,Mn)As DFS, the investigation on co-doping effect in (In,Mn)As is still very limited and new phenomena are worth to investigating.”
Comment 2: The paper is unbalanced between materials characterizations and performance.
Answer: One figure (Fig. 2) is about the structural characterizations and two figures (Figs. 3 and 4) are about the properties. Therefore the so-called unbalance may be not objective.
Comment 3: The method section should be detailed in such a way that one could reproduce these results with fidelity.
Answer: The content of sample preparation in the method section is enough for reproducing the results by readers. They are just basic parameters for the field of implantation.
Comment 4: The references list is not updated with the most recent and most relevant work in the field.
Answer: Although the paper does not cite the most relevant papers in DMSs, it has involved newest papers about the co-doping effect in DMSs. They are enough for introducing and stressing the background and significance of this work.
Comment 5: What is the results fluctuation? This should be detailed in the paper. This is mandatory point.
Answer: All experimental data are measured directly from equipment, eg. SQUID or RBS, therefore they are objective. The fluctuation has been calibrated by the machine itself, so we think it is not necessary here.
Comment 6: Discussion of results is weakly supported but data.
Answer: In the discussion section, the data description is always accompanied with the analysis and discussion. It has been well shown in the manuscript.
Round 2
Reviewer 1 Report
All questions were answered appropriately and the menuscript was also modified.
Author Response
Thanks very much for the acceptance from the reviewer.
Reviewer 2 Report
None of the points raised by the referee have been addressed properly. In its current form, I recommend the rejection of this MS.
Author Response
Comment 1: The paper is unbalanced between materials characterizations and performance.
Answer: Thanks for pointing this. We have added one paragraph and one figure of Raman spectra to analyze the materials characterization to make a balance. The paragraph has been marked as red in the revised manuscript.
Comment 2: The method section should be detailed in such a way that one could reproduce these results with fidelity.
Answer: Several detailed conditions of implantation have been added into the experimental section to ensure the reproduction of readers. The added content has been marked red for convenient reading.
Comment 3: The references list is not updated with the most recent and most relevant work in the field.
Answer: According to the comments from the referee, several fresh references in the field have been added.
Comment 4: What is the results fluctuation? This should be detailed in the paper. This is mandatory point.
Answer: Thanks for the suggestion from the referee. Indeed, as said by the referee, result fluctuation is quite important to show the objective of measured data. However, herein all experimental data are measured directly from equipment, eg. SQUID or RBS, therefore they are objective enough. The fluctuation has been calibrated by the equipment itself, so it is not necessary here. Moreover, in the field most of works do not show such a result fluctuation of SQUID measurement [Maciej Sawicki et al., Experimental probing of the interplay between ferromagnetism and localization in (Ga, Mn)As, Nat. Phys. 6, 22-25 (2010); Lin Chen et al., Electric-Field Modulation of Damping Constant in a Ferromagnetic Semiconductor (Ga,Mn)As, Phys. Rev. Lett., 115, 057204 (2015)].
Comment 5: Discussion of results is weakly supported but data.
Answer: Actually, to support the discussion, one aspect is the directly measured data which is exactly what the referee said. Another important part is the comparison between the current work and previous reported to comparably analyze the phenomenon. Both of above two aspects are shown in the manuscript. Therefore, it is enough to support the discussion part.
In summary, we have point-to-point answered questions and made corrections according to the editor and reviewers’ comments. We are now submitting the revised manuscript and we greatly appreciate your kind consideration of its publication in Materials.